# Wikipedia as a tool for contemporary history of science: A case study on CRISPR

Omer Benjakob[1,2‡], Olha Guley[1,2], Jean-Marc Sevin [1,2], Leo Blondel [1,2], Ariane Augustoni[1,2], Matthieu Collet[1,2], Louise Jouveshomme[1,2], Roy Amit[3], Ariel Linder[1,2], Rona Aviram [1,2‡]*

1 System Engineering and Evolution Dynamics, Inserm, Université Paris Cité, Paris, France, 2 Learning Planet Institute, Paris, France, 3 Bezalel Academy of Arts and Design, Jerusalem, Israel

‡ OB and RA are joint senior authors on this work.
* anorona@gmail.com

**Data Availability Statement:** All relevant data are within the manuscript and its Supporting Information files.

**Funding:** Thanks to the Bettencourt Schueller Foundation long term partnership, this work was

## Abstract

Rapid developments and methodological divides hinder the study of how scientific knowledge accumulates, consolidates and transfers to the public sphere. Our work proposes using Wikipedia, the online encyclopedia, as a historiographical source for contemporary science. We chose the high-profile field of gene editing as our test case, performing a historical analysis of the English-language Wikipedia articles on CRISPR. Using a mixed-method approach, we qualitatively and quantitatively analyzed the CRISPR article's text, sections and references, alongside 50 affiliated articles. These, we found, documented the CRISPR field's maturation from a fundamental scientific discovery to a biotechnological revolution with vast social and cultural implications. We developed automated tools to support such research and demonstrated its applicability to two other scientific fields–coronavirus and circadian clocks. Our method utilizes Wikipedia as a digital and free archive, showing it can document the incremental growth of knowledge and the manner scientific research accumulates and translates into public discourse. Using Wikipedia in this manner compliments and overcomes some issues with contemporary histories and can also augment existing bibliometric research.

## Introduction

In recent years, the historically qualitative field of history of science has undergone a data revolution [1], with research increasingly making more use of big data and computational techniques for historical ends [2]. Alongside the rise of digital humanities, a divide has persisted between quantitative historical research and textually rich qualitative work. The result is a historiographic lacuna [3] and a debate regarding the materials and methods which can be researched [4, 5]. Here, we suggest a new resource can be utilized for the context of history of contemporary science by systematizing existing research methods on an unlikely arena that is rich in both bibliometric data and historical text: Wikipedia.

Wikipedia's volunteer-run-editorial process was long lambasted as inherently unreliable. When in 2005 Wikipedia was found by a study by *Nature*'s news site to be as reliable as Encyclopedia Britannica—the news was met with surprise [6, 7]. Issues of accuracy in the encyclopedia anyone could edit, along with "edit wars" between volatile unknown editors, all became

partly supported by the LPI Research Fellowship, Université de Paris, INSERM U1284, to RAv and OB. RAv's work was supported in part at the Technion by a fellowship of "The Israel Academy of Science and Humanities". In either case, the funders had no role in study design, data collection and analysis, decision to publish, or preparation of the manuscript.

**Competing interests:** The authors have declared that no competing interests exist.

a topic of research [7–9]. In recent years however, this narrative has reversed: both academic research and the media have praised Wikipedia's coverage and reliance on sources, which in some cases have been found to be in lock step with science [7, 10, 11], especially in regards to the COVID-19 pandemic [12, 13].

If Wikipedia is indeed meetings its self-defined goal of representing the scientific consensus on scientific knowledge while making use of scientific sources, then we wanted see if this was also true historically: Can Wikipedia articles document shifts in science *over time*? Using Wikipedia's edit history, we hypothesized, could allow to see changes in how a field was represented in the past and allow the tracking of citations, with new ones being added and old ones removed as new knowledge accumulated, sparking reassessments of existing paradigms. Our first study showed how two Wikipedia articles on the Nobel Prize-winning field of circadian clocks managed to accurately reflect the field for over 15 years, even as it underwent scientific revolutions [14]. Others have used Wikipedia in a similar manner to focus on documentation of political events such as the 2011 Egyptian Revolution [15].

Wikipedia provides a rich, open, and accessible source of information as past versions of all articles can be viewed through what is termed the changelog. This continuum of text throughout time compliments the traditional historical practice of textual analyses, yet in this case the reading is of changing versions of the same text as opposed to comparing different scientific reviews and papers. For us, this feature raised the possibility to map the changes of specific parts of the article's text, structure and references and easily track new additions and deletions.

The study of academic sources and publications are mainstays of the history of science. Both qualitative and quantitative researchers make use of them—be it for bibliometric analysis or thick description [4, 16].

The historical methods born with historian Derek J. de Solla Price that made use of publication data [17] joined the works of earlier historians like Robert K. Merton that laid the historiographic framework for research into the scientific revolution [18]. Later on, sociological works, written by historians like Robert Darnton on the history of books, offered a qualitative detail-rich chronicle of the rise of scientific media during the Enlightenment, substantiating the scientometrics of history with rich detail [19, 20].

Unlike academic publications focused on the state-of-art of the field or review papers coverage of the aforementioned, Wikipedia does not aim to publish original research—it only reflects the scientific consensus based on already published sources. In that, Wikipedia also provides a textually rich base for qualitative work regarding narrativization and the communication of science outwards from academia to the public. Importantly, policies enforced by Wikipedia's editors require "verifiable" sources to back all factual claims [21], and every article has a reference list. A small but growing body of bibliometric research based on Wikipedia has also emerged [22, 23] and even found that on medical [24] and science [8] topics English-language articles have an explicit bias towards using academic sources.

Wikipedia thus easily lends itself to such research, providing both data and text that can be used for historical analysis. In this work, we demonstrate this through a case study on the CRISPR field.

In a relatively short period of time, CRISPR-based gene-editing tools have been labeled the scientific "breakthrough" of the 21st century [25]. While CRISPRs were identified in the 1980's, and received their name in 2002 [26], their function remained unclear until 2005, when different labs deduced from *in silico* studies that CRISPR sequences were part of a bacterial adaptive immune system [27–29]. The academic studies that first performed CRISPR-based directed gene editing *in vitro* were famously published in 2012: First from the labs of Jennifer Doudna and Emmanuelle Charpentier [30] and shortly after in a paper of the Virginijus Šikšnys group [31]. These were rapidly followed by publications in February 2013 that

performed genetic engineering *in vivo* in mammals, led by scientists Fang Zhang [32] and George Church [33]. Thus, the field matured from a basic science discovery into the ability to utilize CRISPR-associated proteins like Cas9 for genetic engineering, currently used by countless labs around the globe [34]. Doudna and Charpentier were awarded the 2020 Nobel Prize for Chemistry for their scientific contribution to genetic editing technologies, showcasing how the so-called CRISPR revolution played out over the past 20 years. Told as such, CRISPR's history can seemingly be deduced through academic publications. However, the science itself does not tell the full scientific story.

In contrast to many other groundbreaking scientific discoveries which remain known only within scientific circles, gene editing has also been in the spotlight of much public debate. For example, many news outlets have dedicated reports to developments in the field and debated the ethical implications of "designer babies" [35]. Netflix has even broadcasted a documentary on CRISPR, underscoring its iconic status in popular culture. By now, CRISPR is not a purely scientific phenomenon. Wikipedia, a popular source read and compiled by the general public, strives to document these facets as well.

The CRISPR field's brief history has been riddled with controversies, and legal battles over credit and CRISPR patents were all covered extensively in the media [36]. Most famously, Eric Lander's perspective in Cell, the "Heroes of CRISPR" [37], was met with fierce criticism [38]. Critics claimed that the text offered a biased version of the field's history that minimized the roles of some scientists as part of the patent war raging between academic institutions [39]—going as far as to label Lander the "villain" of CRISPR [40]. This controversy underscores how scientific outlets, even those famous for publishing novel scientific research, may not necessarily serve as reliable historical sources on contemporary science itself.

The encyclopedia's text and sources can thus be viewed as an inclusive media, one that can potentially help track the interaction with additional fields and allow a better understanding of how scientific knowledge ramifies well outside the realm of academic publications.

CRISPR is a prime example of a scientific field that has undergone massive growth during Wikipedia's lifespan. It is an ideal case study as its history is short (i.e., parallel to Wikipedia's lifetime) and multi-faceted: a highly scientific topic with wide-ranging technological and social ramifications. These facets, we found, were documented on Wikipedia and its different articles, supported by scientific, public and popular sources alike. Together, our findings—based on an analysis of the CRISPR article and 50 others with related content—suggest that Wikipedia can indeed serve as a tool in the digital history of contemporary science. To that end, we put forward a methodology and provide automated tools utilizing Wikipedia's data—its articles, their edit histories and their references. Our method relies on both quantitative and qualitative analyses that may help consolidate research into Wikipedia and help address the aforementioned conflict between data and content-dependent historical research.

## Methods & results

### Delineating the research scope

Historical research requires a clear delineation of the field being studied, for instance gathering a collection of academic publications [5]. Similarly, the manner in which a scientific field is represented on Wikipedia requires clear delineation of scope and span—i.e., the articles that touch on it and the time frame being examined. While a single article can provide a rich source of textual and historical data, related articles may represent more nuanced facets of a field—like scientists' biographies or related events and technologies. Identifying these requires sieving through Wikipedia's massive body of articles—currently numbering well above 6 million in English alone.

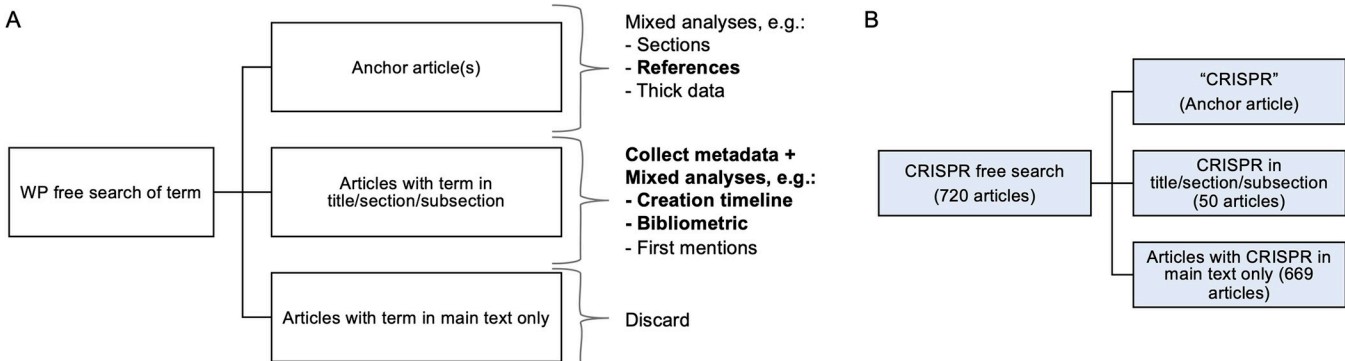

**Fig 1. Workflow for using Wikipedia to research the history of a specific field.** A) Scheme of proposed research flow as supported by our tool: A free search of Wikipedia's English-language articles is conducted to identify relevant articles; these are then filtered to include only those with the term in either their title or that of a section. Next, different analyses can be performed on the anchor article and corpus. Of the listed examples, in bold are the data provided by our tool, the rest are currently collected manually. B) Breakdown of flow scheme in the CRISPR case study, as of June 2022.

For this aim, we propose a stepwise strategy for defining a research corpus about a certain topic. The first step utilizes Wikipedia's free-text search function to find all articles that contain the topic being researched (Fig 1A). In the present study, searching for "CRISPR" yielded 720 Wikipedia articles containing that term, as of June 2022 (Fig 1B). Reading of these articles revealed that the majority made only minor or incidental use of CRISPR. Thus, to permit qualitative analyses on a more focused pool, we designed the second stage of the research funnel, which calls for retaining only those articles with the term in either their title or one of their sections, as these will likely contain a substantial amount of information directly related to the subject. To facilitate these steps of search and filtration we designed a tool to do this automatically for any given term of interest (WikiCorpusBuilder). Continuing with the term "CRISPR", this filtering yielded 51 articles (S1 Table). Out of these, 10 had CRISPR in their title, and another 41 that only had it in the title of one of their sections.

Even among this list of corpus articles, a clear hierarchy arises between those which in subject, text and focus are fully aligned with the topic being researched (which we term the "anchor article/s"); and "auxiliary" articles, those that represent secondary aspects of the topic or instances in which it is embedded within other fields. The distinction is important as it allows us to focus the qualitative work described in the following sections. Here, the anchor article selected was "CRISPR", which was determined semantically based on its title and content.

Within this CRISPR corpus, several auxiliary articles focused on scientific topics, for example the article for "CRISPR Activation", "Cas9", or "CRISPR gene editing"; while others had wider scientific topics, such as "Antibiotic", "Gene knockdown", and "Genome editing". Also included were articles with broad topics, for example "Wheat" which had a section on CRISPR-edited strains of grain. Another group of articles were those dedicated to scientists, like the 2020 Nobel laureates Doudna and Charpentier, awarded for their groundbreaking work in the field; or Šikšnys, who also played a pivotal role in CRISPR's history. Other science-adjacent articles touched on more social facets of CRISPR, e.g., "The CRISPR Journal" and "Designer baby", showing how cultural aspects are also captured by this method. We therefore concluded that these articles provide a good sample of CRISPR related knowledge.

Another advantage Wikipedia provides is open access to these articles' data, which we harness using our tool to further characterize the corpus. For example, "CRISPR" ranked amongst the top five articles in terms of size, number of references, and number of edits (S1 Fig).

Thus, we have composed a clearly defined research corpus regarding our key term of interest. The corpus can be depicted through the titles (qualitative) or the data (quantitative), and further mixed analyses can take place as will be demonstrated below.

## Mixed method analyses for understanding historical growth of knowledge

After having established the research scope, created our corpus and identified the anchor article(s) within it, we move to the analysis phase. We deployed three different complementary analyses: (1) A qualitative reading of anchor article(s) text and structure, both its current and past versions; a quantitative analysis of the (2) anchor article and (3) of the corpus. All three are based on data and materials readily available on Wikipedia and many aspects of the quantitative analyses have been automated through our tool (Fig 1). The following describes how these were deployed on CRISPR.

Mixed-methods research [41] combines quantitative and qualitative analyses and served as the basis for this research. We employed this in what can be termed Wikipedia-focused "thick big data" [42] (as opposed to content-agnostic big data approaches) which meshes the world of thick description and data analysis. In our case, Wikipedia articles, their edit histories and sources are treated as an initial dataset, which are then analyzed semantically as well as through quantitative methods and then interpreted in a detailed manner.

First, using Wikipedia's "view history" tool, available at the top right of every article (Fig 2A), we can access the anchor article's past versions and perform a comparative reading. Here, we used annual intervals to sample textual changes—at times narrowing the time frame to provide a more detailed account of the article's historical textual growth.

Comparing the article's past versions provided rich historical context: The article for CRISPR was created in June 2005, as what is termed a "stub"—a short entry in need of further elaboration (Fig 2B). This first version included but a single paragraph elucidating the CRISPR acronym and describing the genetic locus. At the time, there was no mention of its relation to bacterial immunity or gene editing, two points which would be integral to the field, and as a result be highlighted in the article's lead text in future versions (Fig 2C).

Next, we augmented this form of textual comparison with structural analyses of the CRISPR article's architecture, i.e., its table of contents, a basic feature of all Wikipedia's articles. This "table of contents" or "section" analysis is done as a mixed method: Quantitatively, we measured the overall number of sections and subsections (Fig 3A); qualitatively, we reviewed their titles and documented the changes they underwent to provide insight into the content of the article, with the section titles serving as a proxy for new units of CRISPR-related knowledge (Fig 3B and S2 Table). Due to the semantic variation between sections, we prefer to gather this data and perform the comparison manually.

As we shall see, the section analysis is intertwined with shifts in the corpus. To understand the historical processes that took place in the CRISPR corpus, we can examine the articles based on their Date Of Birth (DOB), (Fig 3C and S1 Table). Even more so than the appearance of a new section—opening new articles on Wikipedia requires the topic at hand to have a certain level of "notability" [9], and we therefore considered the creation of a new article as an indication that a critical abundance of knowledge and editor interest has been reached. Here too, we combined a quantitative evaluation of the number of articles being created with a content-dependent reading of their titles (Fig 3C and S1 Table). Finally, a side-by-side view of these two timelines (sections and DOB) adds another layer of information, interpreted to provide a narrative to contextualize the findings, as described below.

Qualitative reading of the section titles showed that the structural changes were directly linked to shifts in the article's content, pertaining to either the accumulation of new knowledge

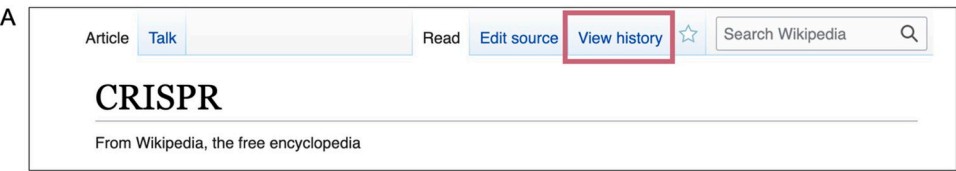

A) An example of the top of a Wikipedia article, note the 'View history'(frame added) tab that enables accessing older versions of the text.

**B** 30/6/2005

CRISPR are direct repeats found in the DNA of many bacteria and archaea. The name is an acronym for clustered regularly interspaced short palindromic repeats. These repeats range in size from 21 to 37 base pairs. They are separated by spacers of similar length. Spacers are usually unique in a genome. Different strains of the same species of bacterium can often be differentiated according to differences in the spacers in their CRISPR arrays, a technique called spoligotyping.

**C** 6/7/2022

**CRISPR** (/ˈkrɪspər/) (an acronym for **clustered regularly interspaced short palindromic repeats**) is a family of DNA sequences found in the genomes of prokaryotic organisms such as bacteria and archaea.[2] These sequences are derived from DNA fragments of bacteriophages that had previously infected the prokaryote. They are used to detect and destroy DNA from similar bacteriophages during subsequent infections. Hence these sequences play a key role in the antiviral (i.e. anti-phage) defense system of prokaryotes and provide a form of acquired immunity.[2][3][4][5] CRISPR are found in approximately 50% of sequenced bacterial genomes and nearly 90% of sequenced archaea.[6]

Cas9 (or "CRISPR-associated protein 9") is an enzyme that uses CRISPR sequences as a guide to recognize and cleave specific strands of DNA that are complementary to the CRISPR sequence. Cas9 enzymes together with CRISPR sequences form the basis of a technology known as CRISPR-Cas9 that can be used to edit genes within organisms.[8][9] This editing process has a wide variety of applications including basic biological research, development of biotechnological products, and treatment of diseases.[10][11] The development of the CRISPR-Cas9 genome editing technique was recognized by the Nobel Prize in Chemistry in 2020 which was awarded to Emmanuelle Charpentier and Jennifer Doudna.[12][13]

**Fig 2. Comparing versions of the CRISPR article.** A) An example of the top of a Wikipedia article, note the 'View history'(frame added) tab that enables accessing older versions of the text. Snapshots from the Wikipedia archive of the CRISPR article: B) the full text of the article when it first opened on June 30th 2005, and C) extract of the lead section's opening paragraphs, as of July 6th, 2022.

or the restructuring of the growing field's representation on Wikipedia. For example, the first sections added in 2010 were "CRISPR Mechanism", "CRISPR Spacer and Repeats," "CAS Genes" and the reference section (Fig 3B and S2 Table). These sections pertain to CRISPR's genetic makeup, and can be collectively referred to as the basic science behind CRISPR.

In 2011, a "Discovery of CRISPR" section was added to the article, which was later renamed "History". The addition of an explicitly historical section in the article indicated a new phase in the scientific narrative it put forward, perhaps the result of a new self-consciousness or understanding that the emerging field was now old enough to have a history of its own. After a few months, a section termed "Evolutionary significance and possible applications" was created. For the next three years it included three proposed applications:

- *"Artificial immunization against phage by introduction of engineered CRISPR loci in industri- ally important bacteria, including those used in food production and large-scale fermentations.*

- *Knockdown of endogenous genes by transformation with a plasmid which contains a CRISPR area with a spacer, which inhibits a target gene.*

- *Discrimination of different bacterial strains by comparison of CRISPR spacer sequences (spoligotyping)"*

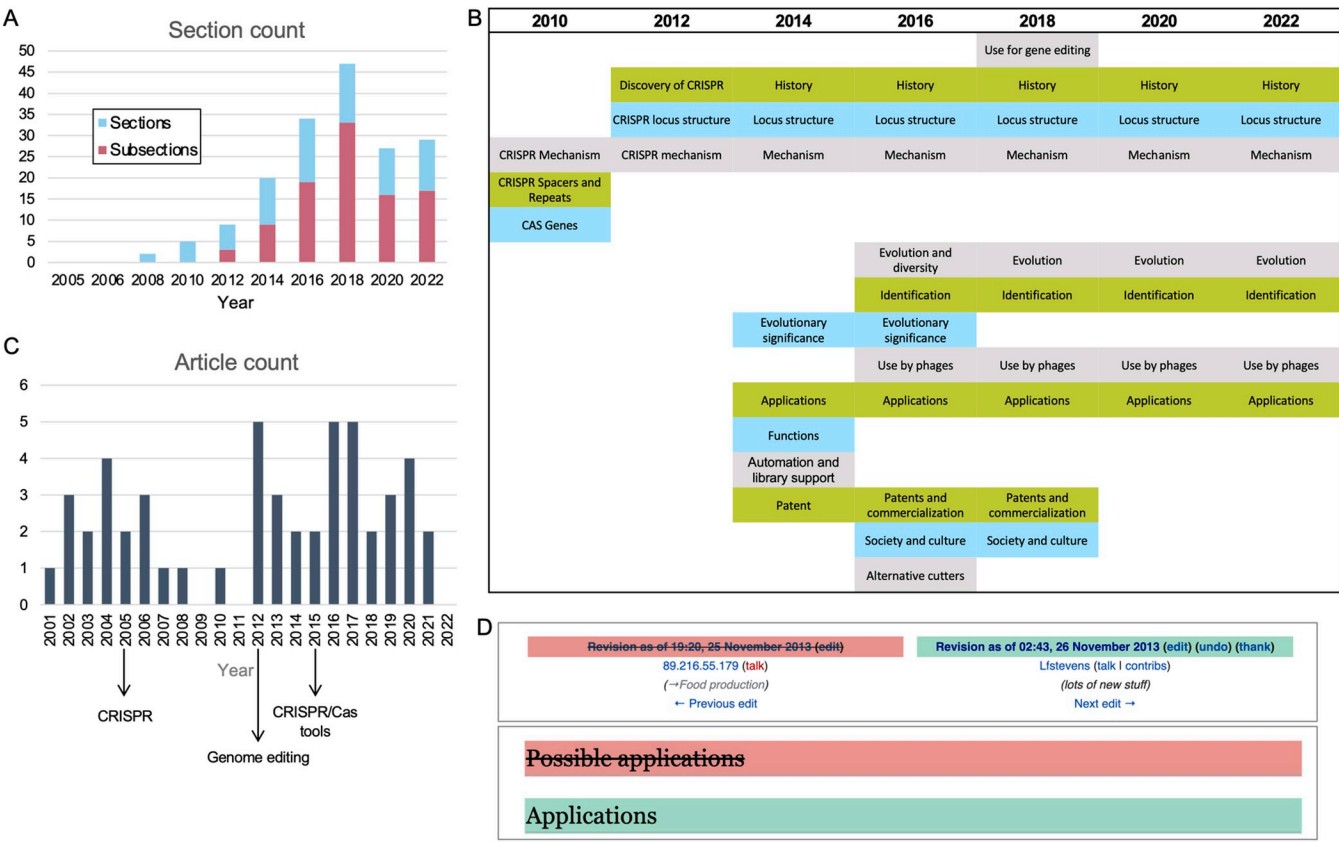

**Fig 3. Growth of CRISPR on Wikipedia—anchor article and corpus.** A) The number of sections and subsections in the CRISPR article since it opened in 2005. B) Titles of the article's sections throughout 2010–2022, sampled biannually. Subsections and those listing sources were removed for clarity and can be found in S2 Table. Alignment and coloring were added manually to highlight sections repeating in consecutive revisions. C) Timeline of the number of the corpus' articles opened each year since Wikipedia was launched (2001). The articles titles and DOB can be found in S1 Table. D) Changelog as of November 25th, 2013, documenting the section title change from "Possible applications" to "Applications" (Other changes that occurred as part of that edit were removed for visibility, and can be found in the archive). All analyses shown occurred until June 2022.

However, these would change in the following year. In April 2013, a user called *Genomeengineering* made what would be their sole yet extremely significant contribution to Wikipedia: Adding the 2012 paper by Doudna and Charpentier, and the two 2013 publications by Zhang and Church. They also amended the list of possible applications so it now included "genome engineering at cellular or organismic level by reprogramming of a CRISPR-Cas system to achieve RNA-guided genome engineering". In November of that year the section's title changed from "Possible applications" to "Applications". This serves as a prime example of how the article documented changes in the field as they took place, with Wikipedia's native "View history" tool's textual comparison function offering snapshots of the "revolution" (Fig 3D).

Alongside this section's growth, which also saw the birth of the "Further reading" section, and a section dedicated to "External links" was expanded, providing access to new utilities developed for CRISPR researchers. For example, a link to a "comprehensive software" for CRISPR guideRNA design was added as well as a link to a tool "for finding CRISPR targets."

At the corpus level, this period also saw a spurt in article creation, with a number of CRISPR-related articles being created, like "CRISPR interference". At this time, more articles directly based on or linked to CRISPR science and its applications were also created. For example, articles like "Genome editing" (2012) and "Cas9" (2013). It is also during this phase that the articles for scientists linked to its discovery were opened: an article about Doudna was

created in 2012, coinciding with the publication of her landmark *Science* paper [30]. Soon thereafter, articles were created for "Epigenome editing" (2014) and "CRISPR/Cas tools" (2015). Thus, qualitatively, this period can be seen as covering the emergence and establishment of the applicative side of CRISPR.

On March 31, 2014, a few weeks after Doudna and Charpentier applied for a patent for their work, a "Patents" section was opened. In 2016, the section dealing with patents was expanded to include a "Patent and commercialization" subsection that detailed a list of patent holders that at the time were fighting in the courts over legal ownership and in academic media over credit (S3 Table). At the corpus level, we observed the creation of articles for Charpentier (2015) and Šikšnys (2016), in tandem to the credit and patent wars raging over their respective discoveries.

In February 2019, with the patent wars reaching their resolution in the courts, the section (then four paragraphs long) was completely removed from the article. However, it was not deleted, but rather migrated to a new article called "CRISPR gene editing," opened that month in a big text-migration out of the anchor article. Also migrated was the section "Society and culture", which described the ability to conduct human gene editing in terms of the wider social debate about it and the policy changes it sparked, alongside a subsection on "Recognition" that attempted to attribute the CRISPR discovery to specific persons. The migration of key sections into "CRISPR gene editing" is evident in the drop in the number of sections in 2019, alongside the uptick in the number of articles in the corpus like "genome-wide CRISPR-cas9 knockout screens", "the CRISPR Journal" and "LEAPER gene editing" (Fig 3).

This later phase also continued to document the growth of the biotech industry based on CRISPR, for example CRISPR Therapeutics, a company co-founded by Charpentier, received an article in 2021, further highlighting the field's maturation and growth in technology.

Tellingly, 2020 also saw the creation of a "Pandemic prevention" article, which, in tandem with the COVID-19 pandemic, detailed all the medical and scientific attempts to preempt viral outbreaks—including those that could potentially make use of CRISPR. Articles like these raise an interesting question regarding the role of CRISPR in other bodies of knowledge and warrant an examination of the wider corpus.

## Cross-pollination: CRISPR as a body of knowledge

Our analyses thus far shows that knowledge on Wikipedia is rarely confined to a single article, but is rather stored in groups of articles that are constantly changing and cross-pollinate one another. On Wikipedia, this process can take on two distinct forms: new articles opening about the topic that directly address it, or existing articles changing to include new text, references or sections dedicated to the scientific topic's intersection with other bodies of knowledge. Tracking the migration between articles can illuminate how knowledge diffuses.

To better understand the temporal aspect of CRISPR's representation across articles on Wikipedia we next compared the DOB of the different articles in our CRISPR corpus and the date the term CRISPR was first mentioned in them.

Of the 51 articles in the CRISPR corpus, 26 already had the term "CRISPR" in their first version (Fig 4A). Among these were the articles for researchers like Charpentier, Šikšnys and Francis Mojica. This group also included articles for scientific topics discovered in later stages of the CRISPR field's growth, like "Cas12", and articles reflecting CRISPR in culture, like the aforementioned academic journal. With few exceptions, like "CRISPR" and "CRISPR interference", opened in 2005 and 2010, respectively, articles that were created with CRISPR already mentioned in their first version were mostly opened post-2014 (Fig 4B).

The 24 articles that lacked "CRISPR" in their inception provide insight into the growth of the field over time. Importantly, this analysis shows how many concepts now associated with

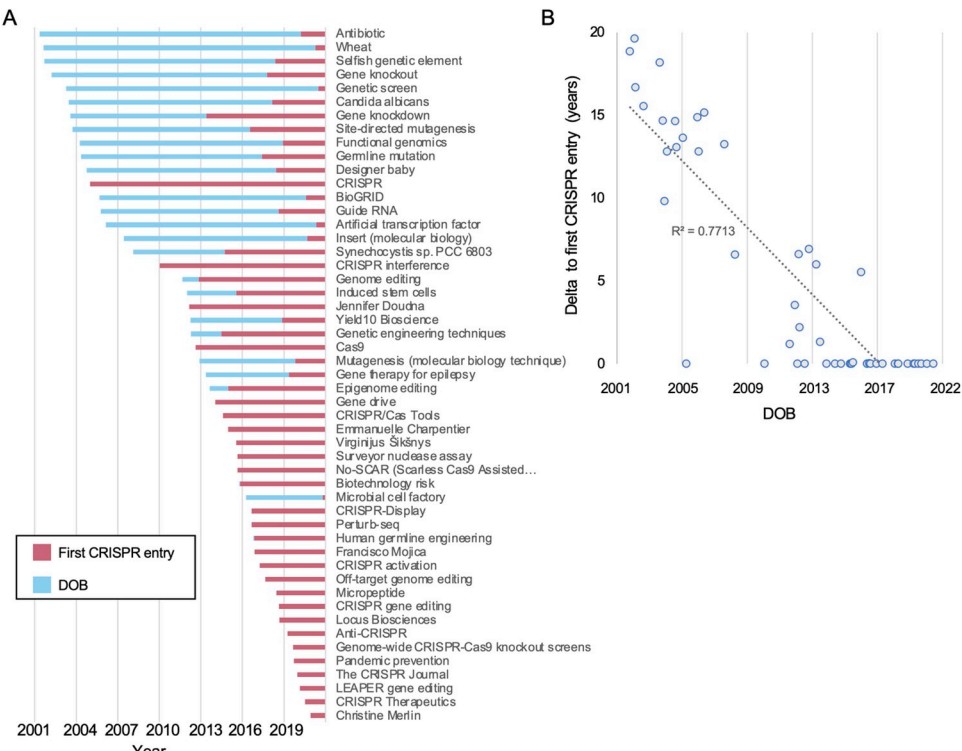

**Fig 4. Comparing an article's creation date and CRISPR's first mentions.** A) An article's date of birth (DOB, blue) compared to the year of it first mentioned "CRISPR" (red), sorted by the former. B) The relation between the DOB and the time it took for the first mention of CRISPR of each article. Displayed is a linear trendline and $R^2$.

CRISPR did actually exist prior to its discovery or its application in gene editing was known. Prime examples are "Gene knockout" and "Gene knockdown", which in fact predate CRISPR. However, as we saw, in a later stage their content was recast to take CRISPR into account and the articles were retroactively affiliated with the CRISPR field (in 2017 and 2013, respectively). Similarly, "Genome editing" was opened in 2012 but mentioned CRISPR only in 2014. The article "Designer baby" opened in 2005 and depicted what was initially only a theoretical issue used in "popular scientific and bioethics literature." However, this changed with CRISPR's rise to prominence and since 2018 it directly referenced CRISPR, with a lengthy debate in wake of the "He Jiankui affair", in which the Chinese scientist created in 2018 the world's first so-called CRISPR babies in a widely reported incident.

We could also observe CRISPR's interface with other scientific fields through articles related to wider topics. For example, the two oldest articles in the corpus, "Wheat" and "Antibiotic", were opened in 2001, and were late to adopt "CRISPR" some twenty years later.

In sum, this analysis revealed a clear divide between articles that mentioned CRISPR from the onset and those that incorporated the term only in later stages: In general, this analysis underscores how CRISPR ramified across Wikipedia not just in the form of new articles, but also recasting older ones.

## From lab to public: Wikipedic bibliometrics map the diffusion of knowledge over time

All claims on Wikipedia need to be attributed to a verifiable source [21]. For our purposes, these references constitute a source rich with text, information and data for additional

analyses: combining quantitative bibliometric analyses like citation count, with content-dependent evaluation of the actual sources, to better understand the types of references supporting the "anchor" article. Quantitatively, we have previously developed two bibliometric analyses for Wikipedia articles—the "SciScore", which gauges the ratio of academic to non-academic sources (ranges 0–1) [12], and the "Latency", which gauges the duration between an academic paper's publication and when it was referenced in a Wikipedia article [14].

Our automated tool scrapes only the reference list of each article in the corpus, which is then further parsed to identify and characterize its different sources: ".org", ".com" and those containing DOIs/PMIDs/PMCs (i.e., scientific papers). Thus, we can assign a SciScore at both the corpus level and that of an individual article.

We found that the CRISPR anchor article was supported by 208 external sources in its "References" and "Further reading" sections (Fig 5A). The article's SciScore was 0.92, ranking 13/51 in the corpus (Fig 5B and S2A Fig). The top cited journal was *Science* (23 papers), followed by *Nature* and *Cell* (14 each), (S2B and S2C Fig). These results are consistent with previous analyses of Wikipedia articles focused on scientific topics that show that these make use of peer reviewed, high-impact factor academic publications [8, 23].

To attain a historical perspective, we next analyzed the temporal aspect of the above discussed bibliometric parameters, which were compared and contextualized to the changes in sections (Fig 3A). We found that these metrics, and overlapping trends between them, served as markers for important events in the history of the field. A prime example of this can be seen in the aforementioned "Patents" section: on March 6, 2014 Doudna's and Charpentier's patent application was published online and a few weeks later the "Patents" section was opened in the CRISPR article (S3 Table). It cited the US Patent Office website. By 2015, after the Broad Institute was awarded its own patent and the appeal against it was filed by the universities representing Doudna and Charpentier, the article's text changed to indicate that, "As of December 2014, patent rights to CRISPR were still developing." The text also noted that there was "a bitter fight over the patents for CRISPR", a claim supported by this new type of citation which grew increasingly present in the CRISPR article: non-academic sources, in the form of both news articles about the legal cases and even the patents themselves. For example, the claim about the "bitter" legal battle was sourced to a story in MIT Technology Review, a popular science news site, while also referring directly to specific patents and or formal application documents made public online. Overall, the section included a laundry list of patent holders and claimants with a hodgepodge of popular and legal sources as citations. Throughout its entire existence, all the sources in this section were non-academic.

The fact that non-academic sources were deployed in the article to support non-academic aspects of the CRISPR history shows how these types of sources can document non-scientific ramifications of scientific developments. However, the entrance of non-academic sources was not limited to patent debates and also touched on CRISPR's growing social prominence. For example, the 2015 selection of CRISPR as "Breakthrough of the year" [43] was supported by links to popular media sources. Together with the patent links, these non-academic sources led to a decrease in the article's SciScore during this phase (Fig 5B).

Collectively, these highlight how bibliometric shifts are reflective of substantive changes in the article's texts, which in turn are reflective of real-world developments in the field, both in terms of the science and of the social debates it inspires.

We next conducted bibliometric analysis on the entire corpus. We found a number of articles with high SciScores (like "CRISPR interference" or "Cas9") alongside those with low percentage of academic sources, like that for Mojica or the concept of designer babies (S2A Fig). This indicates a correlation between the scientificness of an article's topic and its SciScore, with biographical articles for scientists for example, usually ranking lower than those on scientific concepts.

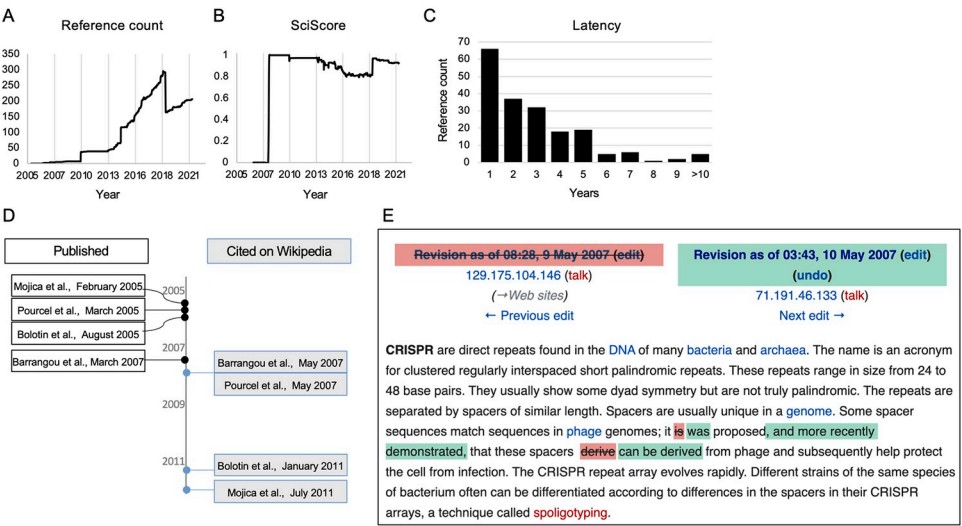

**Fig 5. CRISPR-bibliometrics on Wikipedia.** A) The number of references in the CRISPR article's reference section since it opened until December 2021. B) "CRISPR"s SciScore (shown until December 2021). C) The article's references latency distribution (i.e., duration between a scientific paper's publication and its integration into Wikipedia). D) A timeline comparing the date of selected publications (black frames, left) to their citation in the CRISPR article (blue frames, right). E) A snapshot comparing two versions of the CRISPR article from May 2007, showing how changes to the wording of the text were linked to the citation of Barrangou et al., 2007.

The "CRISPR" article ranked among the top ten in terms of scientificness. To further gauge its current score with the state of the available research, we determined the latency of all the article's references. This analysis revealed a distribution varying between a single day to over 30 years, with a median latency of 1.7 years (Fig 5C). This bibliometric data can be contextualized through the example of the integration dynamics of publications relating CRISPR to bacterial immunity (Fig 5D). Rodolphe Barrangou was the R&D director of genomics at DuPont chemicals manufacturer, who was first to have harnessed CRISPRs to provide immunity for their industrial bacterial strains. The resulting study was published in 2007, and was integrated into Wikipedia that year, a mere two months after going online. In this edit the text changed from "it **is proposed** that these spacers . . . protect the cell from infection" to "it **was** proposed, **and more recently demonstrated**, that these [can. . .] help protect the cell from infection" (bold added), (Fig 5E).

Only after this experimental demonstration were three landmark, yet *in silico*, papers from 2005 added to the article. These three studies which computationally supported the bacterial immune system hypotheses were added with a relatively large latency: Pourcel et al., 2005 was added two years after its publication, while Mojica et al., and Bolotin et al., were added only in 2011—six years after publication. By this time, the text and the early references, as well as CRISPR's function in bacterial immunity, now backed by experimental evidence, were all inserted into the article's lead section, too.

In sum, these quantitative shifts in bibliometrics, we found, were the result of textual changes in the article. This links together our different forms of analyses: the bibliometrics are linked to the historical shifts in the text which together reflected changes in the scientific field itself.

## Quantitative comparison between fields on Wikipedia

We next aimed to examine whether the aforementioned methodology can provide insight into other scientific fields on Wikipedia. Therefore, we deployed our automated tool on two

additional terms, "Circadian" (as in circadian clocks) and "Coronavirus", which we have studied in different manners in earlier works [12, 14] and thus serve as control groups to some degree. We hence created three corpuses side by side, at roughly the same time—June/July 2022, and demonstrated how some of the quantitative analyses described above can be utilized to create comparable yet distinct findings regarding different fields.

As we observed for the CRISPR field, a substantial number of articles can be easily identified and selected to be part of a research corpus—with 51, 138, and 306 articles for "CRISPR", "Circadian", and "Coronavirus", respectively (Fig 6, S4 and S5 Tables). While varying in size, all corpuses are within a range that allows for reading and examination of their titles. Such examination validated that they indeed provide a diverse assortment of articles of different types that are relevant to each field—for example, articles for scientists alongside those for scientific terms or events. For example, the corpus for "Circadian" yielded the articles "Circadian rhythms" and "Sleep", and the corpus for "Coronavirus" yielded articles both about the pandemic like "COVID-19 pandemic in Japan" and more generally for "Virus".

After an initial corpus creation, the first automated analysis generates a timeline based on each articles' DOB. A side-by-side view of all three corpus timelines (Fig 6A) illustrates how different fields display different modes of growth. For example, the "Coronavirus" timeline reveals a clear divide between scientific articles like "Pandemic" (2001) and "Spike protein" (2006), created early on in Wikipedia's history, and post-pandemic articles like "Wuhan Institute of Virology" (2020). This timeline clearly shows how, with the outbreak of the pandemic, articles about the virus ballooned, but also how these were supported by a network of preexisting articles [12]. Meanwhile, the "Circadian" timeline exhibits a seemingly random distribution of article creation, with anchor articles ("Circadian Clock" and "Circadian Rhythms"), and auxiliary articles opening regularly over time. Some DOBs appear to tell a compelling scientific story—e.g., Paul Hardin, first author of the landmark paper highlighted in the 2017 Nobel declaration [44], received an article in 2017 —but these seem anecdotal. Interestingly, the biannual peaks are likely a product of American chronobiologist Eric Herzog's university course [45], which has students contribute to articles of their choice linked to the field. This DOB pattern or lack thereof can be explained by the fact that unlike the timeliness of CRISPR or coronavirus, circadian clocks is a more mature field. As such, its growth, as our previous work has shown, is reflected in a more subtle manner on Wikipedia, with a paradigmatic shift in the field being documented in minute nuanced textual detail [14]. Broadly, this suggests that article creation time is perhaps more applicable for contemporary and what can be termed "active" or even "emerging" fields.

One similarity between all three timelines is an increase in article creation centered around 2005–7, a period which has been shown to have held a massive surge in article creation in Wikipedia in general [46].

Our tool also supports automated scraping of bibliometric data. This analysis showed that the top ten journal references in all three corpuses were dominated with high impact-factor academic peer-reviewed publications (Fig 6B). Alongside prestigious scientific publications like Nature or PNAS, we can observe how each corpus refers to field-specific publications: For example, the Journal of Biological Rhythms in the circadian list, Nature Biotechnology for CRISPR, or The Journal of Virology for coronavirus.

Non-academic references (i.e., websites) were also quite field-specific. As researchers from both the circadian clocks and CRISPR fields were awarded a Nobel Prize, the website for the prestigious award was among the most cited in the respective corpuses (Fig 6C). In addition, the Sleep Foundation website was highly cited in the circadian corpus while three genome focused websites were highly cited in the CRISPR corpus. The International Committee on Taxonomy of Viruses (ICTV), which appears in Wikipedia articles for different variants, was among the top 10 .org sites cited in the coronavirus corpus.

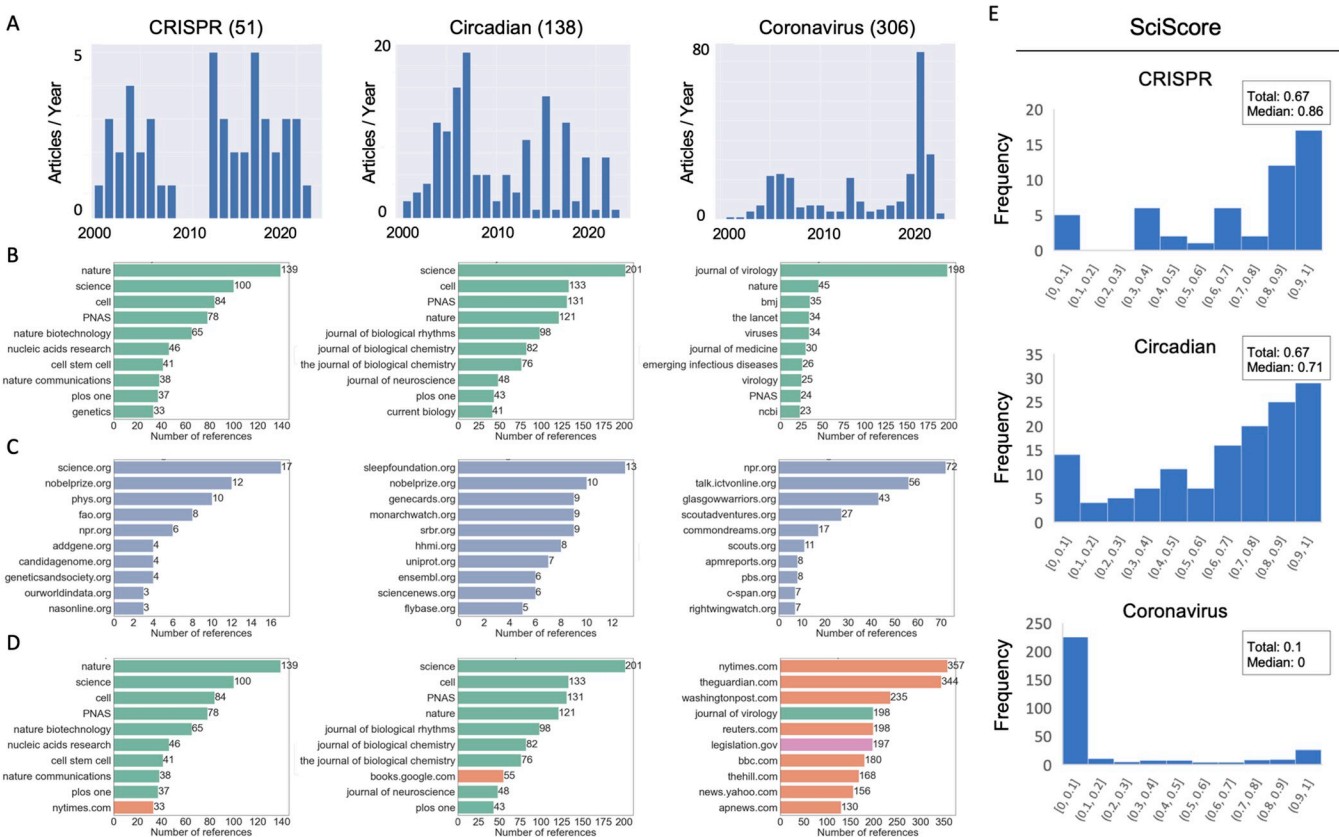

**Fig 6. Comparing Wikipedia corpuses: Different fields show different data.** Corpuses were generated and quantitative metrics automatically collected in June-July 2022, for the terms "CRISPR", "Circadian" and "Coronavirus". The following data are presented: A) the number of articles opened each year, B) the top 10 most cited journals, C) the top 10 most cited.org websites, D) the top 10 most cited references altogether, E) SciScore distribution, along with the total (sum of all references in all articles) and median scores of the articles' distribution.

In general, we observed that the CRISPR and circadian corpuses relied more on scientific literature, while "coronavirus" referenced mostly.com sources (Fig 6D), which is also reflected in the different corpuses' SciScore (Fig 6E). It appears the more prominent a scientific field is societally, the lower its SciScore: for example, the non-scientifically focused CRISPR-corpus article about designer babies which had a relatively low score, as did the circadian-corpus article of "Start school later movement." Meanwhile, the more clearly scientifically focused articles "Surveyor nuclease assay" and "CSNK1D" had high scores. The patterns of SciScore distribution show how different fields manifest differently and that comparing them can shed light, for example, on how much public, as opposed to purely scientific interest, a field has online.

In summary, these analyses show how the same research tools and methods yield very different results for different research fields, all of which can facilitate the initial steps needed towards the creation of future case studies into how scientific knowledge is represented on Wikipedia over time.

## Discussion

Here, we delved into Wikipedia's archives to examine the way a prominent scientific field, CRISPR, was represented from the site's launch in January 2001 until 2022. By reviewing the CRISPR article's history, we saw that the article started off describing the "basic science" behind CRISPR, and was updated in the wake of the publication of canonical works in the

field. Over time, the article grew, and with the emergence of gene editing technology it forked off into a number of affiliated articles with a more narrow focus, while the original CRISPR article offered a consolidated overview of the scientific narrative of CRISPR in bacterial systems. The article's text and its different citations served as a rich record of the growth of academic knowledge, the legal battles CRISPR sparked and the academic credit wars over what the journal *Science* called the "CRISPR Craze" [47], as well as the popular interest in the field.

This case study allowed us to flash out some essential metrics which can be used to conduct similar research, and we thus propose a method that can be deployed in the service of researching the history of contemporary science on other topics using Wikipedia. Automated tools were developed and are openly supplied to support this research permit work on additional topics, though combining these with manual and semantic work are key to contextualizing findings and interpreting them to provide substantial historical insight.

## Using Wikipedia for the history of science

Our findings join a small yet growing body of research dedicated to using Wikipedia for historical purposes. Previously, we analyzed the growth of two Wikipedia articles dedicated to the circadian clock field through their edit histories ("Circadian clocks" and "Circadian rhythms"), using them to ask whether the article's text reflected changes taking place in understanding how biological clocks work [14]. Within that more focused case-study we observed the importance of following the academic references, and developed the Latency metric. Meanwhile, our study on COVID-19 used large-scale quantitative bibliometrics to understand how the pandemic affected large swathes of articles during its "first wave", putting forward metrics such as the SciScore to qualify hundreds of articles based on their reference list [12]. Together, these underscore the key role academic sources play on Wikipedia and serve as a wider proof-of-concept for the quantitative and qualitative underpinnings of this present study.

Wyatt suggested in a theoretical paper that Wikipedia could be used as a primary source in historical research [48]. From the edit history of articles, to metadata for traffic and even talk pages, he envisaged treating the open-source encyclopedia as an "endless palimpsest". This is an idea that has also previously (2010) been expressed as an artwork: "The Iraq War: A Historiography of Wikipedia Changelogs" by artist James Bridle was 12-volume a book comprising all the versions of the article dedicated to the war in Iraq, with the online edit wars serving as a proxy for the real-world conflict. The aforementioned study on the Egyptian protest movement attributed historical significance to the addition of the word "revolution" to Wikipedia articles' titles, taken to be reflective of the real revolution playing out in the streets [15]. This is a captivating demonstration showing the value of attributing historical significance to semantic shifts in Wikipedia articles, in line with our usage of sections and titles.

From the perspective of digital humanities and big data, an algorithmic approach was previously deployed to mine the text of tens of thousands of Wikipedia articles to try to map the history of knowledge since the dawn of human history, using network science and semantic analysis to "put the ideas of Kuhn to the test". The study, currently a preprint [49], makes interesting findings, while highlighting the lack of a unification in methods in current Wikipedia-based historical research. To our knowledge, no academic demonstration nor a clear method has previously been put forward as to how researchers can actually use Wikipedia to utilize its historiographic potential to serve as this "endless palimpsest".

Numerous studies have examined Wikipedia and bibliometrics [23], even those that focus on science [8, 50]; but none that clearly link scientometrics to historical methods [17]. Others from the more humanistic side of academia have worked to connect the digital arena to contemporary fields like discourse analysis, based on the works of Michele Foucault [51].

However, these too are all theoretical works and as of yet no programmatic paper has outlined how Wikipedia can be actually used for historical research - especially not in the interest of following shifts in contemporary science.

Mapping out additional fields through our suggested methods can eventually support theories and models of scientific growth in a resolution never before possible. An initial method for selecting such future case studies could be to focus on the topics selected by Science and others as "Breakthrough of the Year"—these and their relevant Wikipedia articles are documented in a special list on Wikipedia [52] that could serve as the origin of many corpuses. Scientific developments that have garnered public interest over the past two decades, from the human genome project to Alpha Fold, could also serve as lucrative case studies, each providing a unique and rich dataset of text and information that could then be compared.

## The advantages of Wikipedia

Wikipedia easily lends itself to research of this type. A digital and open website that is easily searchable, it also allows open use of its API for more complex queries and even provides a full dump of the entirety of Wikipedia in each language, including articles' full edit history.

Importantly, English Wikipedia's fixed article structure and uniform style allows comparable historical work across different fields, primarily since all articles are structured in a similar way: a lead text, table of contents, sections and then a reference list. This feature, in combination with the convenience of the "View history" function, facilitates in-depth analysis of the same line or section over time (for example, as was done here for "Patents") in a manner difficult to imagine for comparing texts of academic literature. Moreover, cross-analyses of different subjects can yield results comparable through standardized metrics, like the DOB timelines, and the Latency or SciScore. The structural similarity creates a sort of internal control that lays the groundwork for a rigid research system that can be utilized by others and applied to additional fields.

Past versions that did not survive Wikipedia's mob review process or that included facts that were considered true at the time but have since been rendered obsolete prove especially interesting from the perspective of the history of science. For example, with CRISPR, a December 2005 version of the article described *Cas1* as the "most important" of the *Cas* genes, and one that is "present in almost every CRISPR/Cas system." This was more cautiously reworded in July 2010 so that, "The most important of the Cas proteins appears to be Cas1, which is ubiquitous" in CRISPR systems. In March 2011, *Cas1*'s ubiquity was no longer said to be linked to its importance, and for the past decade the article has made due with noting in a subsection dedicated to CRISPR locus that "[m]ost CRISPR-Cas systems have a Cas1 protein." These changes were the result of new knowledge forcing a reevaluation of the preexisting scientific narrative regarding CRISPR: *Cas1* was not falsified per se, rather its importance in CRISPR's story was reassessed. Another example from the CRISPR article can be seen in the shift in section title from "Potential Applications" to "Applications" regarding gene editing (Fig 3D). These are examples of what can be termed "negative" knowledge—knowledge whose relevance was negated by new "positive" discoveries that outweighed it in significance. However, as such, its degradation of scientific status in CRISPR's narrative has much value from the historical perspective. Wikipedia, we suggest, is an inclusive media that documents both positive and negative knowledge—the accumulation and the rejection of scientific facts through its edit history.

Moreover, unlike social media websites that collect user data for financial reasons, posing a privacy threat and creating ethical dilemmas for researchers, Wikipedia collects no such information as it has no such business model. This makes it not only attractive to volunteers willing

to donate hours to writing and editing the site, but also makes Wikipedia and its data ideal material for social research. Wikipedia's texts are not single-handedly written and are edited collectively in a form of what is termed peer-production [53]. Though this system is not without its flaws, in the context of the contemporary history of science it proves a valuable resource: documenting the consensus regarding certain facts and fields' growth in real-time and in potentially minute details.

Additional advantages that Wikipedia offers in respect to bibliometrics are numerous and deserve their own section.

## Wikipedic bibliometrics

Various studies analyzing Wikipedia's references, even those that focus on science [8, 23, 50]; exemplify the use of Wikipedia for bibliometric research, and to a degree support the view that Wikipedia is much more inclusive than academic publications, making use of non-academic sources usually excluded from scientific papers. Here we implicitly study this using our Sci-Score, and contextualize its trends through the historical thick description. On "CRISPR", for example, legal sources or popular media were added to support the "patent war", which was also expressed in a drop in the article's SciScore. The expansion and then contraction of the "Patents" section (S3 Table), in tandem to the patent wars and their resolution in the courts, show how this historical inclusivity touches to both the text and to the sources.

The SciScore reveals a different historical perspective when comparing the CRISPR and Coronavirus corpuses. We previously discovered a decrease in the SciScore as the pandemic grew to public prominence and more articles about it were opened [12]. This was because many of the new articles opened post-pandemic depicted its social ramifications and outcomes, while the pre-pandemic articles focused on the science behind the virus. In the CRISPR anchor article, the SciScore revealed a completely different process: As CRISPR began as a purely scientific discovery, the decrease in SciScore (~2013–2018, Fig 5A) was found to be the result of its growing public prominence outside scientific circles and the appearance of the first non-academic sources about the looming "CRISPR Craze" [47], followed by the much-publicized patent and credit wars, and finally the wider social, ethical and policy debates it sparked —backed by popular yet respectable sources.

Our latency analyses revealed that CRISPR, a nascent field, was making use of extremely up-to-date papers, in some cases references were added within days of their publication. Meanwhile, the circadian clock article had a median latency of five years [14]. This coincides with the respective histories of the fields: CRISPR is a high profile and emerging field, with advances being mirror almost instantaneously on Wikipedia. On the other hand, clocks, which is a more mature field that has been around for decades, was found to be based on contemporary but also older research which predated Wikipedia. Meanwhile, Coronavirus had a major 17-year peak in latency, exactly in line with the SARS pandemic of 2003; showing how research from a preceding viral pandemic provided the backbone of the sourcing for the 2020 pandemic [12]. Together these show how the character of each field is reflected in its bibliometrics.

One hypothesis regarding the potential of the SciScore and Latency is that this dynamic may also be taking place in other articles that began as purely scientific but are increasingly taking on social significance. Tracking articles that have short latencies and high SciScore which then begin to decrease could serve as a method for identifying new fields only now starting to make waves in terms of public interest. In light of emerging attempts to harness Wikipedia for trend detection [54, 55], this idea remains to be examined as more case studies will be created in the future.

Using Wikipedia bibliometrics also has value from the scientometric perspective. Measuring the impact of scientific research is a well-established field that has in recent years expanded

the metrics it works with—no longer just impact factor and citation counting, but also more inclusive metrics like altmetrics. In this sense, Wikipedia, too, can prove a valuable addition in the form of alternative metrics. Asking which papers are cited on Wikipedia and in which context, may provide insight into what parts of academic research are actually reaching the public [7, 56, 57]. As such, our tools can join and enrich existing studies on the history of contemporary science, augmenting work in the field of bibliometrics or even altmetrics, with Wikipedia.

## Limitations

For all its benefits, this method also has its shortcomings. To begin with, corpus delineation can exclude possibly valuable articles—for example, the article for George Church was absent from our corpus despite his seemingly important role in the history of CRISPR. Being a prominent scientist with a broad scope, his Wikipedia article is devoid of the term in any section title, and succinctly mentions his contribution to the CRISPR field under a section titled "Synthetic biology and genome engineering" that despite its topic does not use the key term itself.

From a scientometric perspective, Wikipedia also poses some unique problems: Unlike bibliometric datasets created especially for such purposes, Wikipedia's footnotes are not all properly formatted and issues with their templates exist that make scrapping them consistently hard [58], especially with older articles. Initially, all footnotes on Wikipedia were added manually by editors working directly in wiki-code, the HTML markup language the website uses. Over time, bots and tools were put into place to help this menial task and unify footnotes formatting; in some cases, older articles with older footnotes that did not benefit from this unified new formatting will not be scrapped properly if one uses only Wikipedia's native bibliometric data. To overcome this issue in the present study, we scraped the references from the articles as simple text, regardless of how they were formatted by Wikipedia's volunteer editors. This list of references was then analyzed in search of DOIs/PMIDs/PMCs which were taken as a proxy for academic publications. Nonetheless, other issues exist, for example duplicate DOIs or DOIs included in article's texts and not just as footnotes. A manual validation of our method in random articles revealed this approach had a margin of error that was lower than 5 percent.

Moreover, our method also does not yet address all of Wikipedia's content: Firstly, we only examined English Wikipedia. While it is the largest Wikipedia, and most if not all scientific papers are published in English, language asymmetry has been previously reported across different topics [59, 60]. This is but one of many biases Wikipedia suffers from and including other language editions in future work may reveal different perspectives and richer narratives that are absent from our methods and findings. Even within the English Wikipedia, the talk page, a key arena that is rich in textual data, was not systematically included in this study, though debates about the patent war were found, and these included discussions of which type of sources (legal as opposed to scientific) should be cited on the article in this context.

Another yet untapped facet of Wikipedia touches on visual elements. Wikipedia's sister project, WikiCommons, supports multimedia, usually in the form of copyright-free images, and in this respect we also saw a growth: The first infographic explaining the CRISPR system was introduced to the article in 2009 and the file itself was updated in 2010 to show a more complex understanding of the "CRISPR prokaryotic antiviral defense mechanism", supported by a then-newly published review article [61]. Over time, additional more complex images were added to the article, for example those showing how CRISPR interference could be used for gene editing (S3 Fig). This multimedia aspect can serve in the future as another vector for like-minded research, for example by focusing on how infographics and scientific illustrations document growth of scientific knowledge overtime in visual terms.

We hope our proposed method will encourage use of Wikipedia's ever-changing text as a rich historical source to augment existing work being done in the history of science and contribute to our understanding of the growth of scientific knowledge and its transference to the general public.

## Supporting information

**S1 Fig. The CRISPR corpus in numbers.** The articles included in the corpus, sorted by number of references, size in kilobytes (kB) and number of edits. "CRISPR", highlighted, was among the top 5 articles of each category.
(TIF)

**S2 Fig. CRISPR article's references.** A) The corpus' articles SciScore distribution. B) Peer-reviewed journals cited as references in the article as of June 2022, sorted by the number of references per publication. C) A list of the top cited journals (from B) with ≥5 appearances.
(TIF)

**S3 Fig. Illustrations of the CRISPR model.** Shown are a selection of screen grabs from the CRISPR article, reflecting the evolution of Wikicommons graphics of CRISPR's mechanism of action and key players. These are of different versions of the same illustration (A and B) and of a third illustration added later to the article.
(TIF)

**S1 Table. The CRISPR corpus 2022.** The output of the automated tool for the term "CRISPR", as of 2022-06-27.
(XLSX)

**S2 Table. CRISPR sections.** The "CRISPR" article's table of content was examined at the indicated dates. The number of sections/subsections were counted and appear at the top of each column.
(XLSX)

**S3 Table. Patents section history.** The "CRISPR" article's "Patent" section is displayed for the indicated dates. Separation into different rows in the table were manually done for visibility.
(XLSX)

**S4 Table. The circadian corpus 2022.** The output of the automated tool for the term "circadian", as of 2022-07-14.
(XLSX)

**S5 Table. The coronavirus corpus 022.** The output of the automated tool for the term "coronavirus", as of 2022-07-14.
(XLSX)

## Acknowledgments

We want to thank Dusan Misevic, Bastian Greshake Tzovaras, Marc Santolini, Mad Price Ball, Alex Webb, Gal Manella and all those who provided feedback.

### Code accessibility

Our code for the corpus builder can be found at: https://github.com/RonaTheBrave/WikiCorpusBuilder.

## Author Contributions

**Conceptualization:** Omer Benjakob, Rona Aviram.

**Data curation:** Omer Benjakob, Olha Guley, Rona Aviram.

**Formal analysis:** Omer Benjakob, Rona Aviram.

**Funding acquisition:** Omer Benjakob, Ariel Linder, Rona Aviram.

**Investigation:** Omer Benjakob, Rona Aviram.

**Methodology:** Omer Benjakob, Ariel Linder, Rona Aviram.

**Project administration:** Omer Benjakob, Ariel Linder, Rona Aviram.

**Resources:** Omer Benjakob, Rona Aviram.

**Software:** Omer Benjakob, Jean-Marc Sevin, Leo Blondel, Ariane Augustoni, Matthieu Collet, Louise Jouveshomme, Roy Amit, Rona Aviram.

**Supervision:** Omer Benjakob, Rona Aviram.

**Validation:** Omer Benjakob, Rona Aviram.

**Visualization:** Omer Benjakob, Rona Aviram.

**Writing – original draft:** Omer Benjakob, Ariel Linder, Rona Aviram.

**Writing – review & editing:** Omer Benjakob, Rona Aviram.

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
