## [Decision Letter · Decision Letter 0]

18 Jul 2023

PONE-D-23-06451Wikipedia as a tool for contemporary history of science: A case study on CRISPRPLOS ONE

Dear Dr. aviram,

Thank you for submitting your manuscript to PLOS ONE. After careful consideration, we feel that it has merit but does not fully meet PLOS ONE’s publication criteria as it currently stands. Therefore, we invite you to submit a revised version of the manuscript that addresses the points raised during the review process.

We look forward to receiving your revised manuscript.

Kind regards,

Claire Seungeun Lee

Academic Editor

PLOS ONE

Journal Requirements:

“Thanks to the Bettencourt Schueller Foundation long term partnership, this work was partly supported by the LPI Research Fellowship, Université de Paris, INSERM U1284, to RAv and OB.”

Reviewers' comments:

Reviewer's Responses to Questions

**Comments to the Author**

1. Is the manuscript technically sound, and do the data support the conclusions?

Reviewer #1: Yes

2. Has the statistical analysis been performed appropriately and rigorously? 

Reviewer #1: Yes

3. Have the authors made all data underlying the findings in their manuscript fully available?

Reviewer #1: Yes

4. Is the manuscript presented in an intelligible fashion and written in standard English?

Reviewer #1: Yes

5. Review Comments to the Author

Reviewer #1: This is an interesting paper, with a solid, mixed-method based approach.

For some reason, the references are tangled up (each is in its own brackets, instead of integrated) - I'm assuming your reference manager software got a hiccup? I suggest fixing it in your next revision.

Other than this minor comment, I don't really have much to suggest. One small thing, perhaps: very often people write about Wikipedia as if it was one entity, but there are about 300 different language editions. In fact, if you check studies on cultural diversity of quality of information on different Wikipedias, it turns out that the standards for knowledge quality, presentation, as well as procedures differs significantly. For that reason it may be useful to add a sentence or two mentioning this caveat.

6. PLOS authors have the option to publish the peer review history of their article (what does this mean?). If published, this will include your full peer review and any attached files.

Reviewer #1: No

---

## [Author Response · Author response to Decision Letter 0]

26 Jul 2023

Response to reviewers

We humbly resubmit for publication our manuscript, entitled “Wikipedia as a tool for

contemporary history of science: A case study on CRISPR”.

Please see our point by points response:

Reviewer #1: 

This is an interesting paper, with a solid, mixed-method based approach.

For some reason, the references are tangled up (each is in its own brackets, instead of integrated) - I'm assuming your reference manager software got a hiccup? I suggest fixing it in your next revision.

We apologize for this bug and have corrected the above-mentioned references, as well as reformatted the entire reference section based on PLOS guidelines.

Other than this minor comment, I don't really have much to suggest. One small thing, perhaps: very often people write about Wikipedia as if it was one entity, but there are about 300 different language editions. In fact, if you check studies on cultural diversity of quality of information on different Wikipedias, it turns out that the standards for knowledge quality, presentation, as well as procedures differs significantly. For that reason it may be useful to add a sentence or two mentioning this caveat.

We thank the reviewer for this suggestion, and have amended our discussion with respect to this caveat. The revised text appears in Track Changes, and includes the following lines (and references below):

“Moreover, our method also does not yet address all of Wikipedia’s content: Firstly, we only examined English Wikipedia. While it is the largest Wikipedia, and most if not all scientific papers are published in English, language asymmetry has been previously reported across different topics [59,60]. This is but one of many biases Wikipedia suffers from and including other language editions in future work may reveal different perspectives and richer narratives that are absent from our methods and findings.”

Respectfully yours,

Dr. Rona Aviram and the team

New references:

[59] Roy D, Bhatia S, Jain P. A Topic-Aligned Multilingual Corpus of Wikipedia Articles for Studying Information Asymmetry in Low Resource Languages. Proc. Twelfth Lang. Resour. Eval. Conf., Marseille, France: European Language Resources Association; 2020, p. 2373–80.

[60] Lewoniewski W, Węcel K, Abramowicz W. Quality and Importance of Wikipedia Articles in Different Languages. In: Dregvaite G, Damasevicius R, editors. Inf. Softw. Technol., vol. 639, Cham: Springer International Publishing; 2016, p. 613–24. https://doi.org/10.1007/978-3-319-46254-7_50.

---

## [Editor Report · Decision Letter 1]

17 Aug 2023

Wikipedia as a tool for contemporary history of science: A case study on CRISPR

PONE-D-23-06451R1

Dear Dr. aviram,

We’re pleased to inform you that your manuscript has been judged scientifically suitable for publication and will be formally accepted for publication once it meets all outstanding technical requirements.

Kind regards,

Claire Seungeun Lee

Academic Editor

PLOS ONE

Additional Editor Comments (optional):

Please make sure all the suggested changes in your final manuscript.
---

## [Editor Report · Acceptance letter]

22 Aug 2023

PONE-D-23-06451R1 

Wikipedia as a tool for contemporary history of science: A case study on CRISPR 

Dear Dr. Aviram:

I'm pleased to inform you that your manuscript has been deemed suitable for publication in PLOS ONE. Congratulations! Your manuscript is now with our production department. 

Kind regards, 

on behalf of

Dr. Claire Seungeun Lee 

Academic Editor

PLOS ONE